# Involuntary Separations: Catholic Wives, Imprisoned Husbands, and State Authority

**Susan M. Cogan**

Department of History, Utah State University, Logan, UT 84322, USA; susan.cogan@usu.edu

**Abstract:** In the 1580s and 1590s, the English state required that all subjects of the crown attend the Protestant state church. Those who refused (called recusants) faced imprisonment as part of the government's attempt to bring them into religious conformity. Those imprisonments forced involuntary marital separation onto Catholic couples, the result of which was to disrupt traditional gender roles within Catholic households. Separated wives increasingly fulfilled the work their husbands performed in addition to their own responsibilities as the matriarch of a landed estate. Gentlewomen were practiced at estate business since they worked in partnership with their husbands, but a spouse's imprisonment often meant that wives wrote more petitions and settled more legal and financial matters than they did when their husbands were at liberty. The state also imprisoned Catholic wives who undermined the religious conformity of their families and communities. Spousal imprisonment deprived couples of conjugal rights and spousal support and emphasized the state's power to interfere in marital relationships in early modern England.

**Keywords:** women; marriage; separation; alimony; Catholic; early modern; England

## 1. Introduction

In the 1580s and 1590s, the English state required that all subjects of the crown attend the Protestant state church. Those who refused (called recusants) faced imprisonment as part of the government's attempt to bring them into religious conformity. Those imprisonments forced involuntary marital separation onto Catholic couples, the result of which was to disrupt traditional gender roles within Catholic households. Separated wives increasingly fulfilled the work their husbands performed in addition to their own responsibilities as the matriarch of a landed estate. Gentlewomen were practiced at estate business since they worked in partnership with their husbands, but a spouse's imprisonment often meant that wives wrote more petitions and settled more legal and financial matters than they did when their husbands were at liberty. The state also imprisoned Catholic wives who undermined the religious conformity of their families and communities. Spousal imprisonment deprived couples of conjugal rights and spousal support and emphasized the state's power to interfere in marital relationships in early modern England. This article asks how women experienced separations imposed by the state and what the material consequences of those separations were.

When the English state imprisoned married English Catholics, officials imposed an involuntary marital separation onto Catholic couples. Throughout this essay, "involuntary separation" is usually used to define marriages wherein the couple did not seek marital separation. Marital separation does not seem to have been an official policy of the state's desire to control religious misbehavior. With rare exceptions, government documents do not mention separation as part of a conscious strategy to disrupt the marriages of post-Reformation Catholic couples. In 1585, Thomas Winton of Hampshire suggested to Queen Elizabeth's Principal Secretary, Sir Francis Walsingham, that the most effective way to bring wives into conformity would be to imprison the women and require their husbands to subsidize that imprisonment, "which being addicted to his peny, he utterlie refuseth to doe"

(TNA 12/185, f. 34). If Walsingham took Wilton's advice it was certainly not on a large scale. Overall, involuntary separation was a practice that grew from and responded to state officials' fears about Catholic men's susceptibility to insurrection and about Catholic women's influence over their husbands.

Within a few months of Elizabeth I's accession in November 1558, the English Parliament passed legislation that required Protestant religious practice of all of the queen's subjects. While the queen was unwilling to legislate individual belief, she and her government imposed penalties on religious practice that did not align with the Elizabethan Protestant church. Successive anti-Catholic laws in the 1570s and 1580s placed increasingly stiff penalties on Catholics who continued to practice that religion. The arrival of seminary priests and Jesuits from continental Europe in 1574 and 1580, respectively, undermined the state by making England a Catholic mission field. Ecclesiastical authorities viewed wives as potential missionaries to their husbands from at least the twelfth century; those beliefs about the salvific power of women amplified following the Council of Trent (Cristellon 2012). To make matters worse, Catholic Spain positioned itself as an ally to English Catholics. This potential alliance stoked fears within the English state that if Spain invaded England, English Catholics would fight alongside the Spanish to attempt a coup. Catholic men and women were imprisoned throughout the 1580s and 1590s on a range of charges that included harboring priests, allowing Mass to be performed in their homes, and providing financial support to priests. These same men were imprisoned as a matter of state security when Catholic Spain threatened to invade England, as it did with the Spanish Armada in 1588 and again in the late 1590s. As a result, some Catholic families spent two decades with the male head-of-household in some kind of confinement, either in prison or house arrest.

Throughout the 1580s and into the 1590s, the English Privy Council regularly ordered the imprisonment of Catholic recusant men. These imprisonments imposed involuntary separation on Catholic couples. The 1580s was a decade rife with plots to unseat the monarch. At least two major plots were foiled, the Throckmorton and Babington Plots, respectively, and a number of smaller ones failed to make it beyond the planning stage. Still, Catholic Spain's willingness to support insurrection in England created an undercurrent of tension between Catholic subjects and Protestant officials. As a preventative measure, the English state imprisoned prominent Catholic men whenever they perceived a threat from within or without the realm. This started in earnest in 1581, when the Jesuit Edmund Campion was captured and while under torture named the Catholic families that had hosted him for the past year. The Privy Council quickly ordered the arrest and interrogation of the patriarchs of those families, resulting in the imprisonment of dozens of men and the involuntary separation of numerous couples. In the spring of 1588/9, with reports of the Spanish Armada approaching England, the Privy Council sent orders into the counties with "a list of those Recusants which their Lordships have caused to be sent thither", to be detained at Ely and Banbury (TNA PC 2/15, ff. 186–187; PC 2/15, ff. 199–200; Mattingly 1959). It is difficult to determine the population of Catholic recusants in late-sixteenth and seventeenth-century England. The number of practicing Catholics and converts that Jesuits reported to their superiors in Rome did not align with what government agents knew. Two explanations for this are that Jesuits perhaps exaggerated their tallies and that government officials either did not know the full extent of Catholic households or did not report all of the ones they knew of.

In early modern England, marriage was both a religious and a secular matter. Marriages were sanctified by the church and protected by ecclesiastical and secular laws. In England, although the power of secular bodies grew, marriage remained part of the purview of ecclesiastical courts (Ingram 1987). Statistics on marital breakdown are not possible to glean within the level of the gentry let alone all social strata. Susan Amussen noted that formal suits for separation represented a minority of estranged marriages, since most couples, especially those below the level of the elite, did not pursue formal legal processes if they separated (Amussen 1988). Scholars have noted that suits for separation occurred in

very small numbers and most marriages, even troubled ones, remained intact (Stone 1965; Ozment 1983; Ingram 1987; Heal and Holmes 1995, p. 76).

When marital breakdown did occur, separation and divorce were also both religious and secular matters. The laws of coverture meant that separation was difficult without a husband's assent since by law a husband usually controlled all of a couple's assets. Due to the expense involved this option was usually only available to couples with financial means and even then, a legal separation could take several years to achieve (Stone 1979; Gowing 1998). As Tim Stretton has noted, "A married woman might gain a separation from her husband in the church courts, allowing her to leave the marital home, but in the eyes of the common law the couple's marriage remained valid and the rules of coverture still applied" (Stretton 2007, pp. 20–21). During this period, however, separations increasingly became the purview of the state. Couples could "bypass" the church courts and instead sue each other for separation or divorce in secular courts such as the Court of Requests (Stretton 2007, p. 25). Ultimately, despite the value both church and state placed on marriage as one of the foundations of a stable society, those institutions (perhaps especially the state) were willing to grant separations when marital distance was in the best interests of the couple and the realm. Even so, these separations were not dissolutions, but *a menso et thoro*, which allowed couples a break from cohabitation but not a legal divorce (Amussen 1988; Stretton 2007). Formal suits for separation were usually carried out by couples with means to hire attorneys and pay court fees, in other words, the elite.

In late medieval and early modern England, marital separation provoked anxiety for the couple involved. Couples were especially fearful of marital infidelity; during long periods of separation wives sometimes abandoned the marriage, even going so far as to remarry (Hanawalt 2007, pp. 126–26). Between c. 1580–1640 and especially during the last two decades of the sixteenth century, Catholic couples, especially recusant ones, were subject to involuntary separations coordinated by English government officials. While the practice was not official policy nor enshrined in law codes and was handled on a case-by-case basis, there is some evidence that the state was invested not only in allowing martial separation, but in some cases orchestrating that separation. Occasionally, these separations were the result of officials' concerns about a wife's power to influence her husband's religious views. Involuntary separations placed strain on a couple's relationship, children, and finances. Obstinate Catholics could gain their liberty by promising to conform to the English church and following through in practice, but they claimed that to do so would imperil one's soul. This article asks how women experienced separations imposed by the state and what the material consequences of those separations were.

## 2. Materials and Methods

This study focuses on involuntary marital separation in late-sixteenth and early-seventeenth century England. It analyzes social dynamics associated with marriage and religion, through both social and political lenses, as it asks how married couples negotiated state interference in their marriages. The evidence for this study is drawn from government documents, correspondence, and petitions. Records of the English Privy Council include orders for imprisonment of Catholics, the larger political context which propelled those imprisonments, and the results of petitions submitted for a prisoner's release. Lists of recusants and their disposition, either at liberty or in prison, archived in State Papers identify the imprisoned men (and rarely, women) across the realm. Correspondence and petitions allow for analysis of the strategies used to argue for a prisoner's liberty. Both correspondence and petitions must be considered both in light of the tropes they include and as expressions of a supplicant's real experience. For example, women's claims of difficulty in managing business or legal affairs usually handled by a husband, or emphasizing the detrimental effect on the family of a patriarch's extended absence, were simultaneously rhetorical strategies that drew on socially accepted gender roles and representations of lived experience. By making use of gender stereotypes, women increased

the likelihood that their petitions would succeed and reinforced those stereotypes, further inscribing them into the social and political expectations of their patrons.

As is the case for elite families regardless of religion, the letters and papers of many Catholic families did not survive to the present day. This creates significant gaps in available source material and in turn shapes what we know about the Catholic experience. Some families, such as the Treshams of Rushton, Northamptonshire and the Throckmortons of Coughton, Warwickshire, left abundant records. The collections of other families, such as the Vauxes of Harrowden, Northamptonshire, were lost or destroyed, some by fire and others during the Civil Wars. What we know about those families is gleaned from government documents and any records that ended up in the muniments of other families. Michael Questier's description of the disposition of the muniments of the Brownes of Cowdray, viscounts Montague of Sussex, illustrates how those losses occurred. In 1793, the Browne's seat at Cowdray in West Sussex was destroyed by fire. Although the muniments room was spared from the fire, its contents were not cared for and neighbors and curiosity-seekers carried off piles of documents, including deeds, manuscripts, and correspondence. What remained was ruined by weather and birds (Questier 2006, pp. 9–13). Still other collections, such as those of the Brudenells of Deene, are in private ownership.

In the period under examination here, it was rare for women's correspondence to survive down to the present day. Gentlewomen's letters and papers seem not to have been considered important enough to archive in a family's muniments and are often found as waste paper, used as material for book bindings or as wrappers or binders for documents perceived as worthy of safekeeping (Daybell 2006a, 2012). For example, one of Lady Muriel Tresham's letters to her son-in-law, Sir Thomas Brudenell, now survives as a partial scrap used to bind some of Brudenell's papers. Petitions, although a formal social and political instrument, offer insight into the experience of Catholics in various times, places, and situations (Gregory 2021; Cogan 2021). As James Daybell has demonstrated with his work on women's petitions, we hear women's voices most directly when they wrote their own letters or petitions. Those women were, overall, confident in the authority they wielded in what was a highly political instrument (Daybell 2006b). We need to be alert to how an amanuensis or a spouse writing on a woman's behalf might have obscured her meaning or intent, but we must not dismiss those documents as evidence of her voice.

The extant papers of the Tresham family cover nearly four decades and therefore offer an opportunity to trace a separated couple's strategies over time. Many of Lady Tresham's draft petitions that remained among the Tresham Papers were in Sir Thomas's hand, but we cannot assume that means Muriel had no part in creating these communications. As was typical of some gentlewomen raised in the first half of the sixteenth century, Muriel's handwriting was crude. She could write, but her hand was not the smooth italic style that gentlewomen of the next generation had. She would not have written her own petitions—she did not even write all of her own letters to her husband. Therefore, we cannot assume that an amanuensis was evidence of anything other than someone to write the communications as she directed them. The letters in her hand make clear that she had a strong role and a powerful voice alongside her husband. Furthermore, the petitions she wrote after she was widowed indicate that she was an experienced petitioner. The style and voice of the communications remains constant, which suggests that she had a central role in creating these missives regardless of whose hand wrote them.

### 3. Discussion

#### 3.1. The Agency of Catholic Wives

Scholars have examined the role, agency, and experience of Catholic wives in Reformation England since the publication of Bossy's (1976) *The English Catholic Community*. Bossy argued that recusant Catholicism was matriarchal, since recusant men endangered their status, income, and family property through their religious nonconformity. For most Catholic men, conformity with the English Protestant church was preferable to recusancy, since even perfunctory displays of conformity were usually enough to protect land, title,

and fortune (Walsham 1993). Under English law husbands were not culpable for a wife's criminal behavior, but societal expectations held that a husband should be able to control his wife's behavior, including her religious practice. (Rowlands 1985). This reality complicated gender roles in some Catholic families as women became the primary custodians of Jesuits and seminary priests who ministered to recusant Catholics. Recusant wives claimed enhanced agency through their religious nonconformity, especially as protectors of resident and itinerant priests, and of their own households. The matriarchal character Bossy detected is a direct effect of the involuntary separation of Catholic couples. Women defended their refusal to attend church when required to do so before county or state officials. In short, Catholic women were both subject to early modern patriarchy and actively inverted it as they became the "women on top" that patriarchal society feared (Davis 1974; Rowlands 1985; Dolan 1999). Women like Margaret Sheldon defended their households from intrusion by government agents or nosy neighbors (Enis 2018). Other women, such as the Vaux women, used their social status to intimidate agents who raided their homes, thereby "deflect[ing] pursuivants' zeal" (Lux-Sterritt 2011). Catholic gentlewomen and noblewomen could use their authority and reputation to establish centers of Catholic practice in a community—effectively missionary centers that were known to officials and which were instrumental in converting some of those officials away from Protestantism (Questier 2006; Cristellon 2012). For some couples, adherence to Catholic practice, whether by the wife only or by the couple as a pair, so disrupted usual gender roles that it reordered the marriage contract (McClain 2018). In 1587 officials in Bedfordshire lamented that some of "the most dangerous people to be trusted at libertye" were women "that ar in truth the most willfull and most obstinate recusantes". Despite repeated presentments and indictments for recusancy, these women "persist[ed] . . . to the evill example of others, and great contempte of Hir Highenes lawes . . . " (TNA SP 12/208, f. 22; The Earl of Kent to the Council). Five years later, officials in Dorset and York wrote to the Privy Council to ask how to proceed with recusant wives, especially those women whose husbands conformed (TNA PC 2/20, ff. 26, 428). This question reverberated across the country, not confined to specific counties or regions.

The involuntary separations to which Catholic couples were subjected complicated how gender roles were carried out in practice. The frequent imprisonment of recusant men like Sir William Catesby, Thomas Throckmorton, Sir Thomas Tresham, and Thomas Wilford inverted the gender dynamic in their households and granted recusant wives greater agency than they otherwise would have had. Women's work expanded to include support of and intervention for imprisoned husbands and an enhanced role in protecting family lands (Cogan 2021, pp. 195–96). These involuntary separations were numerically significant. For example, in March 1589/90, the Privy Council ordered the imprisonment of thirty-five recusant men, seventeen in the custody of Richard Arkenstall at the Bishop's Palace in Ely and sixteen in the custody of Richard Fines at Banbury Castle and Fines's house at Broughton in Oxfordshire (*APC* vol. 18, p. 415). Since most of these men were heads-of-household, their imprisonment created hardships for wives and dependents numbering in the hundreds. Their imprisonments not only amplified inversions to gender order in discourse and societal fear, as Fran Dolan has argued, but also in practice as Catholic wives assumed the mantle of the *paterfamilias* (Dolan 1999).

Government officials recognized not just the agency but also the persuasive power wives had over religious belief and practice within their local communities and their households. The Privy Council complained that women were responsible for "corrupting" and "perverting" the religious conformity of their friends, neighbors, servants, families, and children (*APC* vol. 24, pp. 9, 34). In the late sixteenth century, the Jesuit John Gerard and, a century later, Bishop Richard Challoner recorded numerous instances of wives converting their husbands or bringing them back to the Catholic Church and out of conformity with the Elizabethan Protestant Church (Gerard 1951; Challoner 1827; McClain 2018, p. 138). Gerard's and Challoner's accounts align with Privy Council records noting men's recidivism from Protestant back to Catholic practice because of the influence of

their wives, thus confirming that this aspect of women's power was a very real concern for the state. Indeed, this was a practice sanctioned by the Catholic Church as part of its policy established during the Council of Trent. From the mid-sixteenth century through the late-eighteenth century, the Catholic Church recognized and encouraged wives to be missionaries in their own marriages. The Church hoped (or perhaps expected) that Catholic wives would convert 'heretic' spouses, to influence a husband who had fallen astray to renew his commitment to Catholicism. This reality had political implications when applied to diplomatic marriage, such as the union between Prince Charles (later Charles I) and the French princess Henrietta Maria (Freist 2011; Cristellon 2012).

The state used women's religious fervor as a rationale to place additional barriers between already-separated spouses. State papers frequently reference officials' concerns that a wife's influence could discourage a man's conformity to the English church. In the late 1580s and early 1590s, the Privy Council refused to allow Lady Neville and Lady Catesby to visit their imprisoned husbands (TNA PC 2/17, 845; PC 2/17, 847). Indeed, the Privy Council worried about the influence of Catholic gentlewomen more broadly, to the extent that it issued orders to multiple counties to imprison women who encouraged their communities or households into Catholic worship (*APC*, vol. 23, pp. 182, 188, 202–3, 215–16, 228; *APC* vol. 24, pp. 9, 334). These documents reveal inherent contradictions in patriarchal stereotypes: women were weak, but they were powerful enough to draw men into religious nonconformity. Officials responded by revoking wives' permission to visit husbands in prison or limiting the number of days she could stay with him, thereby making more difficult the couple's experience of involuntary separation. The Nevilles were a case in point: in the summer of 1588, coterminous with the Spanish Armada, Lady Neville undermined the state's efforts to bring her husband into conformity with the Protestant church, which prompted the Privy Council to refuse her access to her husband (TNA PC 2/15, f. 264). Yet, neither policy nor state officers were rigid, and officials could reverse course and relax the restrictions they had imposed, probably in response to a change in a couple's obedience to authority. In July 1590, the Privy Council instructed the Lieutenant of the Tower of London to restrict Lady Neville's visits to her husband to one or two per week, a relaxation of the highly restrictive access the couple had previously experienced (TNA PC 2/17, 847).

### 3.2. The Experience of Catholic Wives

Strong wives skilled at estate- and household-management might have inadvertently contributed to lengthy separations. Involuntary separations placed additional domestic and economic responsibilities onto wives as those women took on additional work, including the roles that their husbands usually fulfilled. While modern commentators might recognize the additional power this reality bestowed on Catholic wives, the women themselves might not have perceived this power as they lived through these experiences. Barbara Hanawalt has argued that the work performed by both husbands and wives was critical to "the survival of the household unit" in medieval England (Hanawalt 2007, p. 117). The same is true for the early modern period. A young gentlewoman's education focused on preparing her to run a household. As a wife, she had an extensive slate of work to perform, from overseeing the provisioning of foodstuffs to malt making, cheese making, brewing, the preparation of medicines and the performance of some medical care, the secular and religious education of children, nurturing her own network of friends and patrons, petitioning, and daily religious observances (Fletcher 1995). As Barbara Harris has noted, these tasks were all part of a woman's "career" managing a great house (Harris 2002). When a wife added some of her husband's work to her already full roster of responsibilities she compounded her labor while at the same time she was deprived of spousal contact and support. The Tresham papers offer insight on what some of these additional duties were. On at least one occasion, Lady Tresham wrote to extended family and friends during Sir Thomas's temporary liberty from prison, imploring them not to visit since her husband was under orders to "avoid concourse of friends . . . during his abode in the country neither

meaneth to visit or be visited" (*HMCV*, p. 74). On another occasion, Lady Tresham traveled to a kinsman's estate to resolve a financial dispute on behalf of her husband, only to be turned away by the relative—something that was less likely to happen to a male head-of-household (*HMCV*, p. 83). She kept her imprisoned spouse informed of her labors through frequent correspondence and she transmitted messages from him to people in the county (*HMCV*, p. 84). All of these were additional tasks added to her traditional women's work managing the household.

Separated wives of gentry status and above had personnel to help them with the administration of the estate. Upper-status families typically had stewards to oversee operations on their estates and correspondence indicates that some of an absent husband's duties devolved to the stewards. As one example, while imprisoned in the 1590s, Sir Thomas Tresham maintained regular communication with his stewards at his Rushton and Lyveden estates, as he oversaw building and gardening works on his estates from his prison cell (*HMCV*, p. 134). However, it was wives, not stewards, who visited prisons to discuss estate business and who carried out many of the tasks imprisoned men could not perform. In 1590, Tresham's sister Lady Catesby visited her imprisoned husband, Sir William Catesby, to discuss estate and legal business; six years later, Margaret Throckmorton, sister-in-law to Tresham and Catesby, visited her imprisoned husband at Banbury for the same reason (TNA PC 2/17, f. 697; PC 2/22, f. 150).

The burden of additional labor for Catholic wives was especially pronounced when recusant husbands were held in close imprisonment and not allowed to see or speak with visitors or to have a servant in attendance. Sir Thomas Tresham was in close imprisonment from January through July 1594, during which time Lady Tresham's only access to her husband was through correspondence (*HMCV*, p. 99). Anne, countess of Arundel successfully petitioned Lord Burghley on behalf of her husband after two years of close imprisonment left him "much decayed" (Hatfield House CP 18/17).

Catholic wives substituted for their husbands in estate management (often with the help of a steward) alongside their usual responsibilities of running the household and overseeing education of their children. Men and women petitioned both individually and in tandem to forward their family's interests; it was not unusual for elite wives to participate in business matters as part of their career as matriarch of a landed family. In fact, throughout the early modern period wives continued to do so, regardless of their religion. For example, Lady Peregrina Chaytor fulfilled both her work and her husband's when he was imprisoned for debt in the Fleet prison in 1689. The couple's letters underscored how deeply embedded gender expectations were in their consciousness and "testifies that household management was conceived of as a partnership which they took seriously" (Muldrew 2003, p. 59). In the previous century, recusant husbands and wives appealed to the Privy Council on the man's behalf. In 1590 Thomas Throckmorton sued for liberty, claiming he needed to attend to legal business that he could not perform from prison (TNA PC 2/17, f. 939). Still, responsibility for additional legal, financial, and perhaps also agricultural aspects of estate administration, in addition to women's usual work and the stress of a husband's imprisonment, probably placed Catholic wives under considerable strain.

In 1583, Lady Tresham wrote to her aunt Bridget, Countess of Bedford and also her cousin Mr. Horseman, to ask for their help in moving Sir Thomas's house arrest to their own house at Hoxton, on the outskirts of London. Lady Tresham emphasized the positive effect such a change would have on her husband's health and the benefit of that change to "we, his wife and many children" who would then have their "principal worldly direct[or]" to guide them, which would bring "me and our children exceeding comfort" (*HMCV*, pp. 29–30). These communications, which occurred slightly less than two years into Tresham's imprisonment, accentuated gendered expectations, particularly that the family was incomplete without the paterfamilias at home. Lady Tresham's phrasing alluded to the submissive female posture that was expected of women. This is especially visible in her claim that the family needed the patriarch's direction and guidance, the implication

being that the matriarch's was inadequate. While modern scholars might recognize the gender trope at work in Tresham's plea, we must also acknowledge that her statement also reflected her experience of forced separation from her husband. Husbands' correspondence and petitions made similar entreaties. Sir Thomas Tresham noted that imprisonment and especially close imprisonment not only deprived men of their liberty but also "our wifes by speciall order depriued [of] our companie" (TNA 12/219, f. 138). Petitioners who used gender sterotypes as a linguistic strategy might have increased the chances of success for their petition, but they also reinforced the gender stereotypes necessary for a successful petition in the future.

Over time, correspondence reveals more traditionally male work devolving onto Lady Tresham. For example, in 1591, she wrote to Richard Culpepper about a mortgage on the Tresham lands (*HMCV*, p. 60). While this is something an elite woman might do as part of her larger responsibilities alongside her husband, Lady Tresham seems to have done more of this kind of work during Sir Thomas's imprisonments than she did when he was at liberty. In cases like this one, we see how intertwined a family's credit was with both the husband's credit and the wife's, as Craig Muldrew has noted (Muldrew 2003).

Catholic wives petitioned their patrons and the Privy Council for their husband's release from prison. Those documents offer insight into the experience of the wives and families of imprisoned men. In 1583, Muriel Tresham sought help from her aunt, Bridget Hussey Russell, countess of Bedford, to clarify the terms of her husband's recent release from prison and his bond for good behavior. Muriel pleaded with her aunt that the Treshams would never again have to "live in this thralled sort separated, and our little children . . . deprived of their father's comfort and direction" (*HMCV*, p. 29). In contemporary usage, "thralled" was understood as held captive, held in subjection, or deprived of liberty. Thus, Lady Tresham's own words testify that her experience of involuntary separation equated to a form of captivity for the entire family. Lady Tresham frequently petitioned on her husband's behalf, and probably more often than the drafts that survived among the Tresham and Cecil papers suggest. In a letter to Lord Burghley thanking him for his patronage, Sir Thomas Tresham mentioned that she had "related to me not onlie the honourable furderance your honour therin vowsafed to ys all but moreover howe favourablie your L*ordship* admitted of her womanishe importunacie in oft troublinge you in her husband*es* suite, and howe speciallie y*our* L*ordship* steaded me . . . in the same. (TNA SP 12/219, f. 138).

Catholic wives also petitioned for the release of materials imprisoned men needed for legal or economic purposes. Jane Carter asked for Sir Francis Walsingham's help in obtaining "certaine goodes and bookes" for her husband William Carter, which were locked up when he was arrested (TNA SP 12/206, f. 184). Walsingham agreed, but Owen Hopton, the Lieutenant of the Tower of London, refused to carry out Walsingham's order. After Jane's death, William's mother Agnes renewed Jane's petition, thus taking over from her deceased daughter-in-law the job of petitioning on William's behalf. The Carter example indicates that this type of work was handled not only by wives, but also by other women in a family: in this case a mother, but at other times sisters played significant roles.

Catholic wives sometimes petitioned on behalf of their husbands for the liberty of other prisoners. Mary, baroness Vaux did so in August and September of 1590. For several years, Mary and her husband William had relied on Sir Thomas Tresham's legal advice. In the early 1590s, the Vauxes were enmeshed in a number of financial and land transactions on behalf of their children, including raising the marriage portions for their daughters, settling the inheritance of the barony on Lord Vaux's second son, George, and negotiating with George's wife Elizabeth (Roper) Vaux. William, Lord Vaux was a passive individual and that trait along with his imprisonments for recusancy meant he and his wife needed support from someone with a strong personality and legal expertise. Thus, Mary, Lady Vaux petitioned the Privy Council not only for her husband's liberty but also for the liberty of his legal advisor (TNA PC 2/17, ff. 849, 871, 895). Petitioning was part of an elite woman's

work (Daybell 2006b). The additional petitioning separated wives were compelled to do because of a husband's imprisonment amplified that aspect of their role.

Extant petitions reveal how accomplished many women were at aligning their pleas with culturally valued justifications. Catholic wives invoked specific tropes connected with gender roles, such as how bereft the family was without the patriarch there to guide their daily life or how difficult the additional responsibilities were for a woman to perform. Lady Tresham's invocation of her children being "continually deprived of their father's comfort and direction", mentioned above, and another instance wherein she pleaded for her own relief "and our children exceeding comfort to enjoy" their father's care are examples of her awareness of the gendered expectations of household life (*HMCV*, p. 30). As a mother, Lady Tresham would have overseen the children's upbringing and education (secular and religious), but the lack of a male head-of-household resident in the home disrupted the gender roles within the household and the family. Pleas of poverty were not just tropes, however, and not limited to Catholic women. A century later, some of the same tropes were still in use. Hannah MacDonnell's petition to recover the maintenance that was part of her marriage contract emphasized the helplessness and poverty she and her seven children faced when her husband was attainted (CSPD William and Mary 1689–1690, vol. 7, p. 334). The crown grant that awarded MacDonnell a £300 per annum income justified the award based on the "great necessity" to which she and her seven children were reduced by circumstances beyond their control (CSPD William and Mary 1689–1690, vol. 7, p. 334). Skill at crafting persuasive petitions and submitting them to the right patron was not solely a trait of Catholic women, but of upper-status women in general. In the early seventeenth century, Elizabeth "Bess" Throckmorton Ralegh, one of the Protestant Throckmortons, petitioned King James's private secretary, Robert, earl of Salisbury, beseeching him to help her to recover her widow's portion and relieve her poverty (*HMCS*, p. 84).

Illness was another culturally valued justification for liberty from prison that Catholic couples invoked. In 1583, Margaret Gage of Bentley in Sussex petitioned Sir Francis Walsingham, principal secretary to Queen Elizabeth I, explaining that her husband was "greatlye incombred *with* dyv*er*s infirmities" due to his two-year confinement in the Marshalsea prison (TNA SP 12/159, f. 139). She asked Walsingham to release Gage from prison and transfer him instead into the "Custody of the highe Sherif of Sussex" (TNA SP 12/159, f. 139). That same year, Lady Tresham petitioned for a change in locale of her husband's imprisonment on the grounds that his health suffered in his current situation, subject to a "wayward warden . . . too badly lodged", exposed to smoke and "continual heat ready in this hot, wet season" (*HMCV*, p. 29). In November 1588, Sir Thomas Tresham requested that the Privy Council release him from Ely Castle on the grounds that he had "fallen into some sicknes thorough his restraint of libertie" (TNA PC 2/15, f. 343). Ralph Sheldon, Thomas Throckmorton, and John Talbot also received reprieves from imprisonment based on their wives' petitions about their ill health (TNA PC 2/13, f. 278; *APC*, vol. 15, p. 348; *APC*, vol. 16, p. 389; *APC*, vol. 17, pp. 198–99). Pleas based on illness could work in the opposite direction too. William Shelley gained liberty by claiming the ill health of his wife required his presence at home (*APC*, vol. 14, pp. 125–26).

Catholic wives were imprisoned when officials feared their religious fervor could create disorder in their households or local communities. Even elderly gentlewomen were a concern, especially when they came from established families and held long-standing authority in their communities. In June 1584, Adrian Stokes and Thomas Cave, two of the Justices of the Peace in Leicestershire, recommended that "old M*ist*res Beauemonte" be restrained of her liberty and "frome the access of suche evell disposed p*er*sons whiche are said to resorte to her", one of whom was her daughter-in-law, Mrs Anne Beaumont of Grace Dieu (TNA SP 12/171, f. 99). They asserted that she was "longe accompted a recusante, and a grete favorer of papystes, who in our opynion cannot but be an evell affected subiecte to her ma*jes*tie and an evell member of this Comon Welthe" (TNA SP 12/171, f. 99). Stokes and Cave argued that Beaumont's religious practice itself was not the problem, but the influence she had over people in the household and within her local

community. The following year, Thomas Wilton estimated that there were at least one hundred recusant wives in Hampshire "and in truth I know not one man in this Shire that hath his wife a Recusante that is sound himself" (TNA SP12/185, f. 34). The question of what to do with recusant wives persisted into the next decade. In 1596, Sir Henry Constable successfully petitioned the queen to request that the recusancy case against his wife be suspended while he tried to bring her into conformity (TNA SP 12/256, f. 180).

Sir George Peckham and his wife were both imprisoned for their recusancy: Sir George in the Tower of London and Lady Peckham in the Fleet (TNA PC 2/13, ff. 303–304). Dual imprisonments like the Peckham's were unusual among the gentry. Any petitions Lady Peckham might have written on behalf of her husbands are not extant, but Sir George's are. In February 1580/1, Lady Peckham's liberty came at the suit of her husband, at least in part the Privy Council's reward to Sir George for his new-found religious conformity. Lady Peckham was free to "repaire unto her husbande . . . or to her owne howse at he*r* best liking" (TNA PC 2/13, f. 303). Sir George remained in prison but was granted "libertie of the leades and garden" within the Tower of London, to have visits from his servants "from tyme to tyme to deale with him touching his private affaires, and that Lady Peckham would "be permitted to resorte unto him and to abyde within the Tower . . . to remayne with him at her pleasure" (TNA PC 2/13, f. 304). As this example notes, some wives sought the state's permission to be imprisoned *with* a husband. Couples sometimes needed extended time together to discuss estate or legal business, if not also for intimate relations (especially if she was still in her childbearing years). Further research into dual imprisonments might reveal the state's rationale and imprisoned couples' strategies for release.

The Peckham case and the situation of imprisoned wives in general reveals how contemporary gender roles and the legal status of married women complicated enforcement of religious conformity. Lady Peckham was released with no mention of her conformity or lack thereof, while Sir George remained in prison despite his conformity. Continued imprisonment was probably a consequence of the state's concern that Lady Peckham would influence her spouse back into nonconformity if both of them were released. State officials were reluctant to imprison men for the recusancy of their wives. In September 1592, commissioners for recusancy in Dorset unable to agree on whether recusant wives may be "committed to prison and so severed from their husbandes, and whether their husbandes are by the lawe punishable by any pecuniary paine for that offence of their wives" asked the Privy Council to decide. The Privy Council deferred their decision and granted that in the meantime county officials could "forbear to committ" recusant wives of conforming husbands (TNA PC 2/20, f. 27). The following summer, six gentlemen in York asked the Privy Council to release their wives, who had been in prison for recusancy for fourteen months. The husbands, all conformists, were eager to have their wives at home again and claimed they had "confidence they shalbe hable aswell in reguard of the last statute as of the love and obedience of their wyves to them to worke and induce them by good perswasions and instructions . . . to yeild themselves to conformitie" (TNA PC 2/20, f. 428). This was more than a request to have their wives back at home. Indeed, these men argued for their own manhood by stating that they would be able to exert their authority to bring their wives into religious conformity. The Council could have refused and delivered a blow to the manliness and reputation of the men in question. Such a maneuver might have simultaneously underscored the questionable gender of Catholic men and highlighted the power inversion of Catholic households (Dolan 1999). The Council directed that all of the women should be released to return home with their husbands.

### 3.3. Spousal Maintenance

Spousal maintenance was central to cases of separation and divorce. In late medieval England, separations sanctioned by church courts and secular courts provided for spousal maintenance and were highly variable (Hanawalt 2007; Butler 2013). In the sixteenth century, a separated wife often had to rely on male family members or the courts to arrange for alimony (Stretton 2007). Enforcement of spousal maintenance was difficult in both the

medieval and early modern periods, but recent work by K.J. Kesselring and Tim Stretton makes clear that both ecclesiastical and secular authorities valued alimony as a means of supporting estranged wives (Butler 2005, 2013; Stretton 2007; O'Day 2007; Kesselring and Stretton 2022, pp. 92–96). A man's support of his wife and children was a societal and political expectation across social strata. Quarter Sessions records reveal how that support could devolve to the state or the community in the case of abandonment or widowhood, as it did for Elizabeth Acton of Coleshill, Warwickshire when her husband abandoned their family, and also for Margaret Dowtye of Salford, Warwickshire in 1626 (Ratcliffe and Johnson 1935, p. 25). Gentlewomen typically petitioned the state or patrons well-connected to the central apparatus when they required maintenance. In cases of involuntary separations related to recusancy, the state did not exhibit the same concern over spousal maintenance as it did in other circumstances, such as spousal abandonment.

The state was concerned about spousal maintenance when a husband abandoned his wife. When a man forced separation on his wife through abandonment, the state could intervene, thus underscoring its own power vis a vis marriage. In the first half of the seventeenth century, the crown and state expected husbands to fulfill their duty to financially support their families. In late 1625 or early 1626 Thomas Acton "for some misdemeanour was enforced to fly the country." When he could not be found, support of his wife and children fell to the town of Coleshill, where the family lived, much to the consternation of the town's residents. Town officials sent the family to Kingham in Oxford, shortly after which the Quarter Sessions justices ordered the inhabitants of Coleshill to support the family, thus making the town and its residents a surrogate family patriarch (Ratcliffe and Johnson 1935). Had the Actons been higher in the social hierarchy, Mrs. Acton might have petitioned the state, as Theodosia Tresham did. Theodosia's husband William (the son of Lady Muriel and Sir Thomas) spent most of his adult life in military service in Flanders, voluntarily separated from his wife. In 1638 he reportedly earned "2000*l.* a year by being colonel under the Prince of Orange" (TNA SP 16/392, f. 166). Some officers' wives traveled with their husbands to foreign posts, but others, such as Theodosia Tresham, were effectively abandoned when their husbands worked abroad (Cogan 2021). While William was well-paid for his service in Flanders, he left his wife in London without adequate financial support. In 1638 and 1639, Theodosia submitted at least four petitions to King Charles I, in which she sought to recover her £4000 marriage portion, citing William's neglect (TNA SP 16/392, f. 166; SP 16/395, f. 196; SP 16/408, f. 320; SP 16/439, f. 17). It seems William and Theodosia's marriage had been strained for a while. In 1624/5 William Greene, Elizabeth Saunders, Elizabeth (Lee) Allen, Humphrey Frodsham, and others attempted to blackmail her, alleging that she had committed adultery with Sir Charles Blount (TNA STAC 8/29/10). Whether she was guilty of adultery or the blackmailers were William's friends and supporters is unclear. The king supported her petition and ordered Tresham to provide for his wife. William first ignored the king's commands, then fought them, arguing that her claims were invalid (TNA SP 16/408, f. 320). Records do not indicate whether Theodosia ever received the maintenance the king ordered.

In cases of involuntary separation, the state did not directly express concern that a separated wife possessed the financial means she needed to maintain a household and provide for her children. However, the state's willingness to allow wives to visit imprisoned husbands to discuss estate business and legal matters suggests that officials were aware of and perhaps even sensitive to the financial hardships that could arise from long term incarceration of a family's patriarch. Privy Council records contain numerous examples of separated wives' appeals for relief. These requests were not for cash support but for release from or relaxation of a husband's imprisonment so that he could resume his economic role within the household. Lady Muriel Tresham's efforts to this effect were mentioned above but she was not unusual in her efforts. Her sister, Lady Anne Catesby, and her sister-in-law, Margaret Throckmorton, made similar petitions (*APC* vol. 19, pp. 102–3, 267).

## 4. Conclusions

Involuntary separation disrupted power between individuals, couples, and the state. A Catholic wife's experience of involuntary separation was not one of power, but of additional work and the deprivation of a spouse. Yet, Catholic wives exercised power through their resistance to the state and their work to secure a spouse's liberty. Petitions were political documents, and petitioning was a political activity on behalf of the political alliances that constituted most elite marriages. By placing couples into involuntary separations, the state enhanced the political aspect of those alliances and encouraged women to engage in political activism in the interest of their husbands, households, and families. While the early modern English state did not overtly devise involuntary separations as a strategy to persecute Catholic couples or to undermine marriages, officials like Thomas Winton were cognizant of the persuasive effect that forced separation could product. The state was also sensitive to the hardships for wives that resulted from imprisonment of their husbands, as is borne out by officials, for example Privy Councilors, granting wives access to imprisoned husbands and agreeing to wives' requests to change the terms of a man's imprisonment—from prison to house arrest, for example. At the same time, the state did not seem concerned about an involuntarily separated wife's ability to support herself—probably because of her status as a matriarch of a landed family. Officials were also keenly aware of the power many Catholic wives wielded with their husbands in matters of religion and revoked access when a wife's influence undermined the state's objectives to bring a man into religious conformity, as the examples of Lady Catesby and Lady Neville demonstrated. Further research into involuntary separations should reveal how state-imposed involuntary separations of Catholic couples differed when wives were imprisoned and husbands were the petitioners for relief.

**Funding:** This research received no external funding.

**Institutional Review Board Statement:** Not applicable.

**Informed Consent Statement:** Not applicable.

**Data Availability Statement:** Not applicable.

**Conflicts of Interest:** The author declares no conflict of interest.

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
