# Peer review of "Involuntary Separations: Catholic Wives, Imprisoned Husbands, and State Authority"

_genealogy, doi:10.3390/genealogy6040079_

Round 1

Reviewer 1 Report

This is an interesting and original examination of an important topic that is well deserving of publication.

The only area that requires attention is the treatment of 'coverture'. My advice would be to limit references to this legal term and to fully explain those that occur. At present, in the places I have indicated on the text, the influence of coverture is insufficiently clear (The addition of Frances Dolan's Whores of Babylon to the references would help here.) The article stands up well without requiring assertions about coverture.

The alternative would be to make an analysis of the implications of coverture to enforced separations a central theme of this piece, but that would require significant further research (especially into the legislative history of recusancy laws including petitioning about those laws) and would potentially upset the balance of the article. 

Author Response

Thank you for your careful reading and helpful feedback on this article. As suggested, I limited references to coverture and removed those references in the sections where the reviewer indicated they were more problematic than they were useful. I added references to Fran Dolan's Whores of Babylon (pp. 5, 10) and also to Natalie Zemon Davis. I appreciate the reviewer's copy-editing notes as well.

Reviewer 2 Report

The article deals with the original and interesting topic of involuntary separations of Catholic couples in England, especially in the 1680s and 1690s, due to husbands being arrested for refusing to convert to Protestantism. Such separations meant that wives, in addition to their duties, had to take care of their husbands' affairs. This was not so much an experience of power, but rather an additional workload for the women, who found themselves actually playing the traditional role of the mediator. Through the instrument of the petition, however, the woman in fact performed a political activity, in which she expressed the interests of the political alliances of which her own marriage was an expression.  A further reflection on the linguistic strategy of these petitions would be appropriate here: if it induced women to political activism, to what extent, however, did it make use of the stereotypes (also gender stereotypes) that ensured the success of the petition - and thus reinforced these same stereotypes?

Sometimes women were also arrested: when it was feared that their religious influence could be pernicious to their husbands and society. I think the author could make use here of the reflections of Dagmar Freist, Popery in Perfection. The experience of Catholicism: Henrietta Maria between private practice and public discourses in The Experience of Revolution in Stuart Britain and Ireland, eds Michael J. Braddick and David L. Smith, Cambridge University Press 2011, pp. 33 - 51 and Cecilia Cristellon, 'Unstable and Weak-Minded' or a Missionary? Catholic Women in Mixed Marriages (1563-1798). In: Gender Difference in European Legal Cultures. Historical Perspectives (ed Karin Gottschalk) Stuttgart, Franz Steiner Verlag, pp. 83-93. Both articles can be found online (academia.edu). 

The article is clearly structured and makes use of appropriate documentation. The bibliography on the subject of separation and marriage would probably be too extensive and the decision to limit it is probably in line with the journal's policy of agility. The paragraphs are structured in the comparison with the norms and practices of non-imposed separation and this contributes to the clarity of the article. In the paragraph 'spousal maintenance' the comparison seems a little too forced to me and the extensive description of maintenance obligations in the case of 'normal' separations partly disturbs the fluency of the general text. 

The article is an interesting contribution to the history of marriage, family, separation, religious and gender politics in late 16th century England. 

Author Response

Thank you for your careful reading and helpful feedback on this article. As suggested, I engaged with and referenced the articles by Freist and Cristellon (pp. 2, 5, 6). I briefly touched on linguistic strategy in petitions (p. 4) but did not have the space to develop this as fully as I would have liked to do. I have retained the material on spousal maintenance (in part because of another reviewer's feedback). However, I have tried to make the comparison more smooth, in response to your comments.